# Torrefaction under Different Reaction Atmospheres to Improve the Fuel Properties of Wheat Straw

**Ricardo Torres Ramos** [1], **Benjamín Valdez Salas** [2], **Gisela Montero Alpírez** [2], **Marcos A. Coronado Ortega** [2], **Mario A. Curiel Álvarez** [2], **Olivia Tzintzun Camacho** [1] and **Mary Triny Beleño Cabarcas** [1,*]

1   Institute of Agricultural Sciences, Autonomous University of Baja California,
    Mexicali 21705, Baja California, Mexico; ricardo.torres26@uabc.edu.mx (R.T.R.);
    otzintzun@uabc.edu.mx (O.T.C.)
2   Institute of Engineering, Autonomous University of Baja California, Mexicali 21280, Baja California, Mexico;
    benval@uabc.edu.mx (B.V.S.); gmontero@uabc.edu.mx (G.M.A.); marcos.coronado@uabc.edu.mx (M.A.C.O.);
    mcuriel@uabc.edu.mx (M.A.C.Á.)
*   Correspondence: mary.beleno@uabc.edu.mx

**Abstract:** This study aimed to produce biochar with an energy value in the range of sub-bituminous carbon by investigating the effect of oxidative and non-oxidative torrefaction on the torrefaction yield and fuel properties of wheat straw. Three independent variables were considered at different levels: temperature (230, 255, 280, 305 °C), residence time (20, 40, 60 min), and reaction atmosphere (0, 3, 6 vol% $O_2$; $N_2$ balance); and three dependent variables: mass yield, energy yield, and percentage increase in higher heating value (HHV). The results showed that it is possible to produce a sub-bituminous carbon type C biochar using oxidative torrefaction, significantly reducing time and temperature compared with non-oxidative torrefaction. The optimum torrefaction conditions were 287 °C–20 min–6.0% $O_2$, which increased the HHV of wheat straw from 13.86 to 19.41 MJ kg$^{-1}$. The mass and energy yields were 44.11 and 61.78%, respectively. The physicochemical and fuel properties of the obtained biochar were improved compared with the raw biomass. The atomic O/C ratio was reduced from 1.38 to 0.86. In addition, the hydroxyl groups in the lignocellulosic structure decreased and the hemicellulose content decreased from 26.08% to 1.61%. This improved grindability, thermal stability, porosity, and hydrophobicity.

**Keywords:** agricultural residues; biofuels; oxidative torrefaction; non-oxidative torrefaction; response surface methodology (RSM)

## 1. Introduction

The waste biomass derived from agricultural activities represents an attractive alternative to meet global energy demand. The importance of waste biomass is that it offers the possibility to obtain energy with no net emissions of $CO_2$, contributing to the mitigation of climate change without jeopardizing food security [1]. Biomass is one of the renewable resources with the highest availability on the planet and currently contributes 10% of the world's primary energy supply [2]. In Mexico, biomass contributes 5.17% of the primary energy supply, with sugarcane bagasse, wood, and charcoal being the most used resources [2]. However, it is estimated that Mexico's biomass renewable energy potential is up to 3569 PJ and is equivalent to 46% of the primary energy supply [3].

Mexico's sizeable renewable coal reserves come from the waste generated in agricultural activities currently not participating in national energy production [3,4]. Wheat cultivation is among the most important agricultural activities, with an annual cultivated area of 553,825.4 ha [5]. Wheat crops have a residue generation index of around 7.3 tons ha$^{-1}$ [4]. Therefore, this crop is estimated to generate approximately 38.8 Mt of wheat straw annually, which is considered an herbaceous and fibrous residue [6,7]. This residue is usually burned in the open air, causing severe environmental and public health problems [4]. The

proper final disposal of agricultural residues is an ongoing issue, even in industrialized countries [8].

Waste biomass's main challenges as a primary energy source is its heterogeneity and limitations in its physicochemical and fuel properties [9,10]. Heterogeneity makes it complicated to convert different types of biomass under the same conversion system, while the unfavorable biomass physicochemical properties are related to low energy density, low calorific value, high oxygen content, high hygroscopicity, poor grinding, and low thermal stability [11,12]. These characteristics make it difficult to transport, store, handle, and increase the operating costs for its conversion into energy [1]. On the other hand, co-combustion is the most used technology in biomass energy exploitation. However, most combustion plants have been designed to operate on coal [8]. Therefore, biomass should present physicochemical properties close to coal to avoid further modifications in the conversion systems. The adequacy of biomass properties is essential to diversify its applications at different scales. Biomass with improved properties can be used as fuel for heating systems and domestic boilers that traditionally use coal [13]. It can also be used as a reducing agent in the direct reduction of iron for steelmaking, where coke is traditionally used [14].

In recent years, it has been reported that torrefaction is an effective method to improve the physicochemical properties of biomass. Torrefaction is a thermal pretreatment that ranges between 200 and 310 °C, producing partial decomposition of biomass lignocellulosic materials [15]. This pretreatment is performed in two atmospheres: nitrogen and oxygen, called non-oxidative and oxidative torrefaction, respectively [9]. Non-oxidative torrefaction is a promising technology since torrefied biomass's combustion characteristics and reactivity are similar to sub-bituminous coal [12]. Most research has focused on biomass's non-oxidative torrefaction [9,12]. The results indicate that non-oxidative torrefaction increases fixed carbon and HHV, decreases oxygen and volatile matter content, improves grinding capacity, and increases hydrophobicity [11]. However, some drawbacks are related to the lengthy process duration, the supply of inert gas, and thermal energy, which results in high operating costs [1].

Recently, special attention has been given to oxidative torrefaction to overcome the drawbacks of the non-oxidative process [9]. Torrefaction using mixtures of nitrogen/oxygen in different types of wastes has been reported [1,9,12,16,17]. Previous studies have evidenced that an oxidizing atmosphere accelerates the thermal degradation of structures containing oxygen, increasing the reaction rate and reducing torrefaction time or temperature [1,12]. Oxidative torrefaction could be an appropriate method to optimize the quality of biomass fuels because it increases the HHV, fixed carbon, and elemental carbon, and decreases oxygen content and hygroscopicity, and improves grindability [9,18]. Nevertheless, other works have found that the indiscriminate use of air or other oxidizing agents can promote oxidation reactions of both the volatiles released and on the surface of the biomass, causing the deterioration of its fuel properties [19–21]. It has also been reported that the composition of the gas used to control the reaction atmosphere directly affects the properties of the biomass and defines its subsequent use [1].

Despite the publication of remarkable works on biomass torrefaction, comparative studies of oxidative and non-oxidative torrefaction of wheat straw are still scarce. It is necessary to analyze and compare the effects of oxidative and non-oxidative conditions on the fuel properties of wheat straw thoroughly. Therefore, this study presents how the oxygen concentration in the reaction atmosphere reduces residence time, torrefaction temperature, and the production of solid fuel with similar properties to sub-bituminous coal without compromising the physicochemical properties of wheat straw. The torrefaction process was optimized using response surface methodology (RSM). In addition, a systematic study of the fuel properties of raw and torrefied biomass was conducted. The results provide essential information for a better understanding of the torrefaction processes, contributing to improving this technology and accelerating its scaling up.

## 2. Materials and Methods

### 2.1. Materials

Wheat straw was collected from agricultural fields in the Mexicali Valley in the state of Baja California, Mexico. The samples were manually cut with scissors to a size of 4–5 cm and were washed with deionized water to remove dust. Then, they were dried in an oven at 105 °C for 12 h. Finally, the samples were stored in hermetically sealed bags for later use.

### 2.2. Experimental Design

The experiment was conducted using a mixed-level and $3^2 \times 4$ full factorial design. The independent variables were temperature (230, 255, 280, and 305 °C), residence time (20, 40, and 60 min), and reaction atmosphere (0, 3 y 6% $O_2$; $N_2$ balance). The response variables were mass yield, energy efficiency, and the percentage increase in higher heating value (PI-HHV). The experiments were conducted in duplicate, and the statistical data treatment was analyzed using the Minitab 17.1.0 statistical package. Furthermore, the analysis of variance (ANOVA) was conducted, and the coefficients of the second-degree polynomials for each torrefaction yield were estimated to generate response surface graphs.

### 2.3. Torrefaction Procedure

The experiments were carried out in a cylindrical quartz reactor heated by a tubular furnace, Lindberg/Blue M (Thermo Scientific$^{TM}$, Waltham, MA, USA). The procedure started by adding 25 g of wheat straw, treated according to Section 2.1.

Then, the reactor was placed in a tubular furnace where the reaction atmosphere was conditioned before torrefaction. This conditioning was carried out in two stages. First, nitrogen gas flow was purged at 300 mL min$^{-1}$ for 10 min to remove the oxygen inside the reactor. Second, the supply of nitrogen was suspended and replaced by a mixture of gases that participated in the composition of the reaction atmosphere, according to the experimental design. This mixture was purged at a 300 mL min$^{-1}$ flow rate and continued throughout the torrefaction process. The heating started 5 min after removing the oxygen with the nitrogen gas flow. The temperature increased at a rate of 10 °C min$^{-1}$ until the torrefaction temperature established in the experimental design was reached. The wheat straw samples were maintained at the torrefaction temperature for a residence time specified in the experimental design. Once the residence time was gone, heating was stopped, and the mixture of gases was replaced by nitrogen gas flow at a rate of 300 mL min$^{-1}$ until cooled to room temperature. Finally, the samples were removed from the reactor and stored in hermetically sealed bags for subsequent analysis.

### 2.4. Torrefaction Yield

Three yield indicators were used to evaluate the efficiency of the torrefaction process: mass yield, energy yield, and PI-HHV. The mass yield was used to examine the influence of torrefaction on mass loss, as per Equation (1) [22]. Energy yield represents the percentage of energy remaining in the torrefied biomass concerning the raw biomass, and is determined according to Equation (2) [23]. PI-HHV expresses the percentage increase in the higher heating value of the torrefied biomass compared with the raw biomass, as defined by Equation (3).

$$\text{Mass yield (\%)} = \left[ \frac{W_t}{W_r} \right] \times 100 \tag{1}$$

$$\text{Energy yield (\%)} = \left[ \frac{W_t \cdot HHV_t}{W_r \cdot HHV_r} \right] \times 100 \tag{2}$$

$$\text{PI-HHV (\%)} = \left[ \frac{HHV_t}{HHV_r} - 1 \right] \times 100 \tag{3}$$

where $W_r$ and $W_t$ represent the weight of raw and torrefied biomass, respectively, and $HHV_r$ and $HHV_t$ refer to the higher heating value of the raw and torrefied biomass, respectively.

### 2.5. Torrefaction Optimal Conditions

Optimal conditions are those that allow obtaining a solid biofuel with an energy value in the range of sub-bituminous coal. Therefore, this type of coal reaches an HHV of 19.3–26.7 MJ kg$^{-1}$, according to ASTM D388. Therefore, the optimum value of PI-HHV will correspond to the one that allows reaching a calorific value in this range of values. On the other hand, the mass yield is expected to be greater than or equal to the sum of lignin, cellulose, and ash content, since torrefaction causes the decomposition of hemicellulose, extractable substances, and moisture removal [24,25]. As for the energy yield, the recommendations of other authors were considered, who estimate that the energy yield should be greater than the mass yield by 10% [24–26].

### 2.6. Comparative Analysis

The analyses performed to compare the physicochemical and fuel properties of the raw and torrefied biomass are presented in Figure 1. The comparative analysis considered the torrefied biomass obtained under optimal torrefaction conditions. In addition, some biomass properties were compared with the fuel properties of sub-bituminous coal type A reported in the literature. The raw and torrefied biomass samples were crushed to a size of 0.29–0.25 mm. All analyses were performed in triplicate and were considered reproducible when they had an error of less than 3.0%.

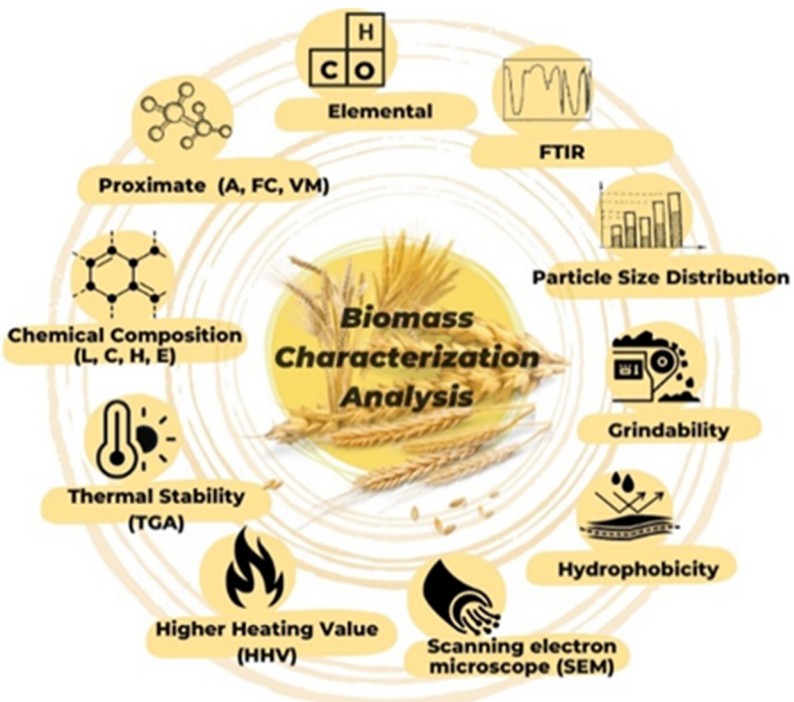

**Figure 1.** Comparative physicochemical analysis.

2.6.1. Proximate Analysis

The proximate analysis included determining moisture content, volatile matter, ash, and, fixed carbon. This analysis was performed by gravimetric methods according to ASTM E870-82.

### 2.6.2. Elemental Analysis

This analysis was used to determine the elemental composition of the biomass as a function of carbon, hydrogen, oxygen, nitrogen, and sulfur concentration. An elemental analyzer, Flash 2000 CHNS/O (Thermo Scientific™, Waltham, MA, USA), was used following ATSM E777-08.

### 2.6.3. Chemical Composition Analysis

Chemical composition analysis was used to determine the components of the lignocellulosic matrix. Each component was determined following a different standard: extractable material (TAPPI T264 and TAPPI 207), lignin (ASTM D1106), cellulose (D1103), and hemicellulose were determined based on difference.

### 2.6.4. Higher Heating Value (HHV)

The energy content of the biomass was determined using the isoperibolic method in a Calorimeter System C2000 (IKA®, Germany) under the ASTM E771-87 standard. The results were expressed in units of MJ kg$^{-1}$.

### 2.6.5. Grindability

The grindability of wheat straw before and after the torrefaction was studied using the Hargrove Grindability Index (HGI). In this test, the sample was ground in a ball mill, PM100 (Retsch®, Haan, Germany), and the percentage of plant material passing through the 75 μm sieve was determined according to the procedure established by Shankar et al. [27]. A calibration curve was performed with the ball mill using coals of different HGI grades (32, 49, 66, and 92).

### 2.6.6. Particle Size Distribution

This analysis was used to evaluate the proportion of fine and coarse particles generated by the grinding of raw and torrefied biomass. The test used 10 g of dry sample with a particle size between 1.41 and 1.68 mm. The sample was ground for 1.5 min and sieved to obtain four fractions with different particle sizes (>1 mm, 0.35$^{-1}$ mm, 0.18–0.35 mm, and <0.18 mm). The particle distribution graph was made from the percentages of biomass retained in each sieve fraction.

### 2.6.7. Hydrophobicity (Moisture Reabsorption)

The moisture reabsorption capacity of biomass (raw and torrefied) was analyzed by immersing 0.5 g dry sample in deionized water at room temperature for 2 h. After this period, the samples were air-dried for 1 h, and, finally, the moisture content in the samples was determined [28].

### 2.6.8. Microscopic Analysis

The morphological changes in the microstructure of the biomass were studied using scanning electron microscopy (SEM) techniques. The analysis was performed in an electron microscope (JEOL JSM-6010LA) with an accelerating voltage of 15 kV. The vacuum pump for the analysis was operated at 50 Pa.

### 2.6.9. Thermal Stability

The thermal stability of the biomass was studied using torrefaction thermogravimetric analysis (TGA) and differential thermal gravimetric analysis (DTG) using a thermogravimetric analyzer, STA 6000 (PerkinElmer, Waltham, MA, USA). The thermal decomposition ranges from 25 to 700 °C, with a heating rate equivalent to 10 °C min$^{-1}$. In addition, nitrogen was used as the reaction gas, with a flow rate of 100 mL min$^{-1}$.

### 2.6.10. FTIR Analysis

The changes in chemical structure caused by torrefaction were studied by FTIR spectrophotometry. A spectrophotometer, Spectrum$^{TM}$ One (PerkinElmer, Waltham, MA, USA), with an attenuated total reflection (ATR) device was used, and spectra were collected at a wavenumber frequency interval of 650–4000 cm$^{-1}$. This technique made it possible to compare the functional groups present in the torrefied biomass with the raw biomass.

## 3. Results

### 3.1. Torrefaction Yield

Table 1 presents the torrefaction yields with the operating conditions. Wheat straw presented its maximum mass and energy yield when the conditions were 230 °C–20 min–0% O$_2$, reaching 82.12% and 88.93%, respectively, whereas the maximum PI-HHV achieved was 49.24% when the conditions were 305 °C–60 min–6% O$_2$. Table 1 displays the effects of oxygen concentration on torrefaction yields. A comparison between the operating conditions 305 °C–60 min–0% O$_2$ and 280 °C–40 min–6% O$_2$ allows observing that both conditions achieved an increase of 36.50% in the HHV while decreasing the reaction temperature and residence time.

**Table 1.** Torrefaction performance of wheat straw.

| O$_2$ | 0% | | | 3% | | | 6% | | |
|---|---|---|---|---|---|---|---|---|---|
| Temperature (°C) | Time (min) | | | | | | | | |
| | 20 | 40 | 60 | 20 | 40 | 60 | 20 | 40 | 60 |
| **Wheat Straw** | | | | | | | | | |
| Solid yield | | | | | | | | | |
| 230 | 82.12 | 78.01 | 76.37 | 73.3 | 68.93 | 62.93 | 69.13 | 63.88 | 55.58 |
| 255 | 64.87 | 62.30 | 59.40 | 58.94 | 53.14 | 50.64 | 54.20 | 52.34 | 45.71 |
| 280 | 48.28 | 47.75 | 46.38 | 47.03 | 45.03 | 43.84 | 46.43 | 43.11 | 41.11 |
| 305 | 43.91 | 43.39 | 42.65 | 43.00 | 41.63 | 40.46 | 41.48 | 38.89 | 37.19 |
| PI-HHV | | | | | | | | | |
| 230 | 8.29 | 10.75 | 11.36 | 11.20 | 11.34 | 15.64 | 10.53 | 12.61 | 16.27 |
| 255 | 21.72 | 22.03 | 26.02 | 22.19 | 23.72 | 29.08 | 24.40 | 27.56 | 33.07 |
| 280 | 32.29 | 32.04 | 33.27 | 33.87 | 35.95 | 37.48 | 36.55 | 40.49 | 42.38 |
| 305 | 36.58 | 38.96 | 40.43 | 40.75 | 43.90 | 46.24 | 44.72 | 47.16 | 49.24 |
| Energy yield | | | | | | | | | |
| 230 | 88.93 | 86.40 | 85.05 | 82.22 | 76.74 | 72.10 | 76.27 | 71.94 | 64.62 |
| 255 | 78.96 | 76.02 | 74.85 | 72.02 | 65.74 | 65.37 | 66.43 | 66.77 | 60.83 |
| 280 | 63.87 | 63.05 | 61.81 | 62.97 | 61.22 | 60.27 | 63.40 | 60.57 | 58.48 |
| 305 | 59.97 | 60.30 | 59.90 | 60.52 | 59.91 | 59.17 | 60.03 | 57.23 | 55.50 |

Second-order mathematical models were adjusted using the Minitab 17 statistical software based on the experimental results derived from the factorial design. The quadratic models (4), (5), and (6) correspond to the mass yield, energy, and PI-HHV, respectively. These models only include statistically significant terms, where X is the reaction atmosphere, Y is the torrefaction temperature, and Z is the residence time.

$$\text{Solid yield}_{WS} = 504.1 - 9.25X - 2.757Y - 0.785Z + 0.1267X^2 + 0.004076Y^2 + 0.02884XY - 0.01759XZ + 0.002639YZ \quad (4)$$

$$\text{Energy yield}_{WS} = 362.1 - 10.17X - 1.714Y - 0.686Z + 0.0023637Y^2 + 0.03310XY - 0.01831XZ + 0.002226YZ \quad (5)$$

$$\text{PI-HHV}_{ws} = -257.7 - 2.55X + 1.735Y + 0.061Z - 0.002519Y^2 - 0.012XY + 0.01204XZ \quad (6)$$

An ANOVA was performed on the quadratic models to evaluate their adequacy and suitability. The analysis of variance results and the summary of adjustment of the proposed models are presented in Table 2. The quadratic models were obtained with a confidence

level of 95% and presented a high significance since the p values were lower than 0.0001. Furthermore, the F values were high, meaning there is only a 0.05% chance that the results of the models are due to noise [8,29]. Previous research has proposed quadratic models for biomass torrefaction with similar statistical criteria [8,29]. On the other hand, it can be observed that the regression coefficients were higher than 0.95 in all the proposed models. A coefficient $\geq 0.95$ is considered an acceptable value since it can explain 95% of the variability of the data [8]. The coefficients obtained were relatively high and are close to the coefficients reported in this type of research [20,30,31].

**Table 2.** ANOVA for the torrefaction yield's response surface models with three predictors: temperature, atmosphere, and time.

| Source | DF | SS | MS | F-Value | Prob. > F | $R^2$-Adjusted |
|---|---|---|---|---|---|---|
| Wheat straw | | | | | | |
| Model Solid Yield | 9 | 5283.34 | 587.04 | 298.22 | <0.0001 | 98.71% |
| Error | 26 | 51.18 | 1.97 | | | |
| Corrected total | 35 | 5334.52 | | | | |
| Model Energy Yield | 9 | 2774.72 | 308.30 | 103.51 | <0.0001 | 96.35% |
| Error | 26 | 77.44 | 2.98 | | | |
| Corrected total | 35 | 2852.16 | | | | |
| Model PI-HHV | 9 | 5286.64 | 587.40 | 538.76 | <0.0001 | 99.28% |
| Error | 26 | 28.35 | 1.09 | | | |
| Corrected total | 35 | 5314.99 | | | | |

Table 3 is obtained from the statistical analysis, which contains the significance level of each term used in the quadratic models. It is observed that the temperature had a more significant impact than the retention time and the reaction atmosphere, and the reaction atmosphere was slightly more important than the retention time.

**Table 3.** Terms used to obtain the mathematical models and their respective *p*–values.

| Biomass | Wheat Straw | | |
|---|---|---|---|
| Indicator | Solid Yield | Energy Yield | PI-HHV |
| term | $p > |t|$ | $p > |t|$ | $p > |t|$ |
| Atmosphere | <0.0001 | <0.0001 | <0.0001 |
| Temperature | <0.0001 | <0.0001 | <0.0001 |
| Time | <0.0001 | <0.0001 | <0.0001 |
| (Atmosphere)$^2$ | <0.0001 | 0.091 | 0.508 |
| (Temperature)$^2$ | 0.030 | <0.0001 | <0.0001 |
| (Time)$^2$ | <0.0001 | 0.852 | 0.268 |
| Atmosphere $\times$ temperature | 0.869 | <0.0001 | <0.0001 |
| Atmosphere $\times$ time | <0.0001 | 0.017 | 0.010 |
| Temperature $\times$ time | 0.006 | 0.002 | 0.535 |

The effects of two significant parameters on torrefaction yields can best be understood in the response surface plots of Figure 2. It is observed that both the mass yield and energy yield rapidly decreased when increasing the reaction temperature and oxygen concentration in the reaction atmosphere. Moreover, an increase in PI-HHV was observed as the temperature and oxygen concentration increased. The effect of oxygen concentration is evident at higher temperatures because oxygen contributes to the surface oxidation of biomass when mass and heat transfer processes are intensified as the temperature increases [26].

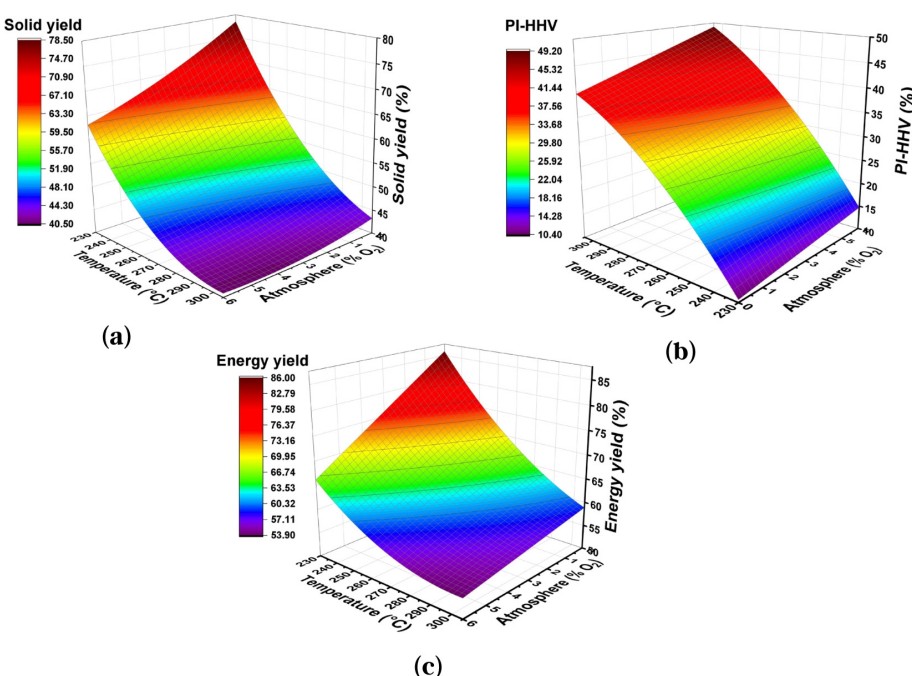

**Figure 2.** Surface graphics of response of temperature vs. atmosphere on the torrefaction yields of the wheat straw for 40 min: (**a**) Solid Yield, (**b**) PI-HHV, (**c**) Energy Yield.

*3.2. Torrefaction Theoric-Optimal Conditions*

The optimal operating conditions for wheat straw torrefaction are established by considering the chemical composition results and HHV analysis, shown in Table 4. The aim is to obtain a solid biofuel with a calorific value in the range of sub-bituminous coal. This type of coal has three subcategories (A, B, and C) that depend mainly on the HHV. Table 5 presents the HHV of sub-bituminous coal for each category and exhibits the PI-HHV required for the wheat straw to reach each subcategory. As for the mass yield, it is considered that the mass loss should not exceed the sum of hemicellulose, moisture, and extractives. Therefore, the mass yield should be ≥58.17%. For energy yield, other authors have suggested that this yield should be approximately 10% higher than the solid yield [24–26]. Considering these investigations, the energy yield of torrefied wheat straw should be ≥68.17%.

Using the response surface methodology, the theoretical operating conditions that allowed obtaining the torrefied biomass in the sub-bituminous coal type C range were 287 °C–20 min–6% $O_2$. Additionally, the mathematical models allowed estimation of the theoretical torrefaction yields for these conditions, as shown in Table 5. According to the models, achieving the sub-bituminous coal type C range is possible with a decrease in the mass yield of 13.7% and the energy yield of 6.06%, considering the mass and energy yields of wheat straw should be higher than 58.17% and 68.17%, respectively. On the other hand, the energy value of sub-bituminous charcoal types A and B were not achieved under the experimental conditions, due to wheat straw having a relatively low calorific value, which requires more severe conditions of temperature, time, and oxidative atmosphere. Therefore, the theoretical conditions and theoretical torrefaction yields for sub-bituminous char types A and B are not presented in Table 5.

**Table 4.** Main properties of raw biomass, torrefied, and coal reference.

| Analysis | Wheat Straw | | Reference Coal [32] |
|---|---|---|---|
| | Raw | Torrefied | |
| Proximate | | | |
| Fixed carbon | 11.45 | 31.48 | 46.77 |
| Volatile matter | 77.63 | 55.92 | 45.53 |
| Ash | 6.34 | 12.60 | 7.7 |
| Elemental | | | |
| Carbon (C) | 39.19 | 50.80 | 70.78 |
| Hydrogen (H) | 5.56 | 4.15 | 6.13 |
| Oxygen (O) | 53.91 | 43.86 | 22.63 |
| Sulfur (S) | 0.20 | 0.32 | 0.44 |
| Nitrogen (N) | 1.14 | 0.87 | 0.02 |
| O/C | 1.376 | 0.863 | 0.320 |
| H/C | 0.142 | 0.082 | 0.086 |
| Chemical | | | |
| Cellulose | 32.51 | 41.84 | |
| Hemicellulose | 26.08 | 1.61 | |
| Lignin | 19.32 | 44.37 | |
| Extractives | 11.19 | 3.59 | |
| Moisture | 4.58 | - | |
| HHV (MJ kg$^{-1}$) | 13.86 | 19.41 | 25.23 |

**Table 5.** Prediction of optimal conditions and torrefaction yields.

| * Coal | * Range HHV (MJ kg$^{-1}$) | ** PI-HHV | Theoretical Conditions (°C-min-$O_2$ %) | Theoretical Yields (%) | | |
|---|---|---|---|---|---|---|
| | | | | PI-HHV | MY | EY |
| Type A | 24.4–26.7 | 76.04–92.64 | - | - | - | - |
| Type B | 22.1–24.4 | 59.45–76.04 | - | - | - | - |
| Type C | 19.3–22.1 | 39.25–59.45 | 287–20–6.0 | 39.25 | 44.47 | 62.11 |

\* Sub-bituminous coal, ** increment needed.

### 3.3. Torrefaction Experimental Optimal Conditions

The wheat straw torrefaction was developed under the optimal theoretical operating condition predicted by the models in Table 5 to obtain biochar in the sub-bituminous coal type C range. The results are presented in Table 6, where it can be observed that, under these operating conditions, the mass and energy yield and PI-HHV were 44.11%, 61.78%, and 40.04%, respectively. A comparison between the experimental torrefaction yields and the yields predicted confirms that the quadratic models adequately describe the torrefaction behavior of wheat straw. On the other hand, Table 6 also presents the torrefaction yields reported in the literature for wheat straw torrefaction. The torrefaction yields obtained in this work are within the range of values reported in the literature. Finally, the biochar produced under these conditions is used in the comparative studies of fuel properties between raw and torrefied biomass.

**Table 6.** Torrefaction yields under optimal conditions and yield reported in the literature.

| Biomass | Torrefaction Conditions | | | Torrefaction Yields (%) | | | Ref |
|---|---|---|---|---|---|---|---|
| | T (°C) | t (min) | Atmosphere | Solid | Energy | PI-HHV | |
| Wheat straw | 287 | 20 | 6% $O_2$ | 44.11 | 61.78 | 40.04 | This work |
| | 300 | 30 | $N_2$ | 46.76 | 64.91 | 38.81 | [33] |
| | 300 | 60 | $N_2$ | 39.10 | 49.90 | 27.00 | [29] |
| | 250 | 360 | $N_2$ | 61.21 | 77.12 | 25.75 | [7] |

*3.4. Comparative Analysis of Fuel Properties*

The fuel properties of torrefied wheat straw were analyzed under optimal torrefaction conditions. The results were compared with the fuel properties of raw biomass and reference type A sub-bituminous coal.

3.4.1. Proximate Analysis

Fixed carbon and volatile matter are essential indicators for defining the quality of solid fuels. A high volatile content indicates a higher fuel reactivity and causes an unstable flame to form during combustion. At the same time, a high fixed carbon content is associated with an increase in energy content and a decrease in fuel reactivity [8]. Wheat straw presented high contents of volatile matter and a low fixed carbon content. However, these values are within the range reported in the literature [7]. After torrefaction, the results of the proximate analysis improved substantially. The fixed carbon content increased to 31.48%, while the volatile content was reduced to 55.92%. The ash content increased from 6.34% to 12.60%. Previous research reported that the torrefaction of wheat straw at 275 °C–30 min produced a reduction in volatile matter content from 74.73% to 50.99% and fixed carbon from 17.13% to 36.46% [33]. The increase in fixed carbon and the reduction in volatile matter improve the physicochemical properties of the fuel. However, the reference coal has a higher fixed carbon content than the torrefied biomass because sub-bituminous coal type A has superior energetic characteristics.

3.4.2. Elemental Analysis

The elemental analysis of the raw and torrefied samples is presented in Table 4. The untreated wheat straw had an unfavorable elemental composition from the energy point of view due to its high oxygen content, and low carbon and hydrogen content. However, after torrefaction, the elemental carbon content increased from 39.19% to 50.80%, while the oxygen content decreased from 53.91% to 43.86%. The energy density of the biomass increased, and its fuel quality improved, which can be observed in the reduction of the atomic O/C and H/C ratios [31,34]. The O/C ratio decreased from 1.376 to 0.863, while the H/C ratio decreased from 0.142 to 0.082. The changes in elemental composition are attributed to the elimination of hydroxyl groups and the release of volatile compounds with high oxygen and hydrogen content, which contribute little to the calorific value of the fuel [35]. As for the elements of environmental concern, the sulfur content remained below the reference sub-bituminous coal. However, the nitrogen content was considerably higher than the reference coal.

3.4.3. Chemical Composition Analysis

Table 1 presents the chemical composition of wheat straw before and after torrefaction. In the raw sample, the most abundant component was cellulose at 32.51%, followed by hemicellulose at 26.08%, and lignin at 19.32%. After torrefaction, hemicellulose suffered the most significant degradation, which was reduced to 2.61%. Hemicellulose gives resistance and toughness to the vegetable structure; therefore, it is expected that the grinding properties of the biomass will improve with torrefaction. In wheat straw, a decrease in the hemicellulose content from 27.78% to 0.34% was reported when the conditions were 300 °C–120 min [36,37].

The percentage of cellulose slightly increased due to the decrease in the other components, reducing from 33.51% to 34.26%. Other researchers reported similar increases from 36.7% to 37.5% when the torrefaction was carried out at 250 °C–2 h [38]. As for the percentage of lignin, a substantial increase was observed after torrefaction. The lignin content reduced from 19.32 to 43.17%. Other research reported an increase in the wheat straw lignin content from 21.28% to 98.40% when torrefaction was performed at 300 °C–2 h [36].

It was expected that there would be a significant reduction in the extractives content as a result of the degradation and volatilization of extractable compounds during torrefaction. However, it was found that torrefied wheat straw contained 7.49% of extractives. The torrefaction effect on the extractive content does not appear to present a clear trend [8]. Some authors have attributed the increase in extractives to residue accumulation produced during the partial degradation of polysaccharides and lignin. Then, these residues are solubilized in the chemical reagents used for extraction tests and are incorrectly reported as extractives [39]. Other authors have reported 0.24% of extractives in torrefied wheat straw samples at 300 °C–2 h and 6.2% in lignocellulosic biomass samples at 300 °C–15 min [36,39].

### 3.4.4. Higher Heating Value (HHV)

Untreated wheat straw presented an HHV equivalent to 13.86 MJ kg$^{-1}$, which is within the range of values reported in the literature [7]. The HHV of wheat straw is similar to the HHV of low-rank coals such as lignite or peat. However, after torrefaction, it increased to 19.41 MJ kg$^{-1}$, corresponding to the sub-bituminous type C coal range. Recent research has reported an increase in the HHV of wheat straw from 18.98 MJ kg$^{-1}$ to 24.32 MJ kg$^{-1}$ when torrefaction was performed at 280 °C–40 min in an inert atmosphere [8]. According to the analyses performed, it was found that torrefaction caused the removal of a higher amount of oxygen compared with carbon [31]. It also contributed to the dehydration and depolymerization of hemicellulose, which resulted in an increase in the content of fixed carbon, elemental carbon, and lignin, which finally led to a rise in the energy content of the biomass.

### 3.4.5. Grinding Analysis

Energy conversion through co-combustion, gasification, and pyrolysis processes commonly requires particle size reduction of solid fuels before being incorporated into the boiler or reactor bed [39–41]. For this reason, grindability is a significant characteristic of solid fuels that allows for estimating the energy consumption associated with particle size reduction [42]. This characteristic influences the buying and selling price of fuels [43]. Currently, the most commonly used analysis to estimate the grindability of solid fuels is the HGI. The higher this index is, the easier the biomass will be to grind and will require less energy consumption to reduce its size [40,42]. It was found that wheat straw has an HGI lower than 32, which indicates poor grinding behavior. After torrefaction under optimal conditions, the grinding capacity of wheat straw improved substantially, reaching an HGI equivalent of 72.6. The improvement in grindability is mainly due to the breaking of the hemicellulose matrix and the depolymerization of cellulose. Other authors reported an increase in the HGI of wheat straw from <32 to 59.8 when torrefaction was performed at 250 °C–2 h in an inert atmosphere [36].

### 3.4.6. Particle Distribution Analysis

Particle distribution analysis is used to understand better the effect of torrefaction on the grindability characteristics of biomass. Figure 3 presents the particle distribution for raw and torrefied wheat straw. It can be observed that 82.54% of particles generated during the grinding of the raw biomass were retained in coarse mesh sieves (>1 mm and 0.35–1 mm), while 17.46% were retained in fine mesh sieves (0.18–0.35 mm and <0.18 mm). After the torrefaction, significant changes in particle size distribution were observed. The percentage of particles retained on the fine sieve (<0.18 mm) increased from 5.39% in the raw samples to 29.33% in the torrefied samples. It was observed that torrefied wheat straw

is more brittle and easier to grind, which are desirable characteristics in solid fuels and reduces energy consumption in grinding. Previous research reported the particle size distribution for raw and torrefied wheat straw grinding at 250 °C–12 h. It was reported that the percentage of particles retained on the fine sieve > 0.25 mm increased after torrefaction from 33.0% to 81.0% [7,44]. Similar studies have reported the particle size distribution for grinding different types of raw and torrefied biomass [7,39,43,44].

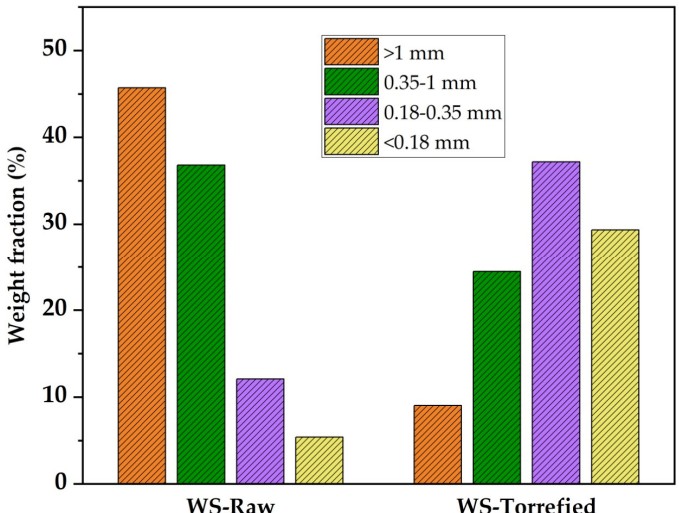

**Figure 3.** Particle size distribution for wheat straw raw and torrefied.

3.4.7. Thermogravimetric Analysis

The thermal stability study was carried out in an inert atmosphere at temperatures between 35 °C and 700 °C. The TGA and DTG curves of the raw and torrefied samples are presented in Figure 4. In both curves, an initial mass loss is observed between 80 °C and 120 °C, attributed to the evaporation of moisture and volatile compounds [45]. The mass loss was less pronounced in the torrefied sample due to the decrease in the hydrophilic character of the biomass after torrefaction [46]. Thermal decomposition continued slowly up to 220 °C in the raw sample and extended to 300 °C in the torrefied sample [39]. The biomass exhibited little chemical reactivity at this stage, but the removal of low molecular weight extractable substances occurred [47].

In the TGA curve of the untreated biomass, a significant mass loss equivalent to 29.98% is observed between 220 °C and 300 °C and is attributed to depolymerization and volatilization of hemicellulose [48]. However, a higher mass loss equal to 57.23% was observed between 300 °C and 390 °C, which is attributed to the depolymerization of cellulose [48]. The next stage occurs between 390 °C and 500 °C and corresponds to the lignin decomposition. On the other hand, in the TGA curve of the torrefied biomass, the decomposition slope is less prolonged than the raw biomass; therefore, the thermal decomposition stages are not easily distinguished. However, it can be observed that thermal decomposition in the torrefied sample starts at higher temperatures, above 325 °C [48]. A comparison between the TGA curves of the raw and torrefied samples allows observation of the improvement in the thermal stability of the biomass after torrefaction. This effect is evidenced by the amount of mass remaining after reaching 700 °C, which increased from 25% in the raw sample to 55% in the torrefied sample.

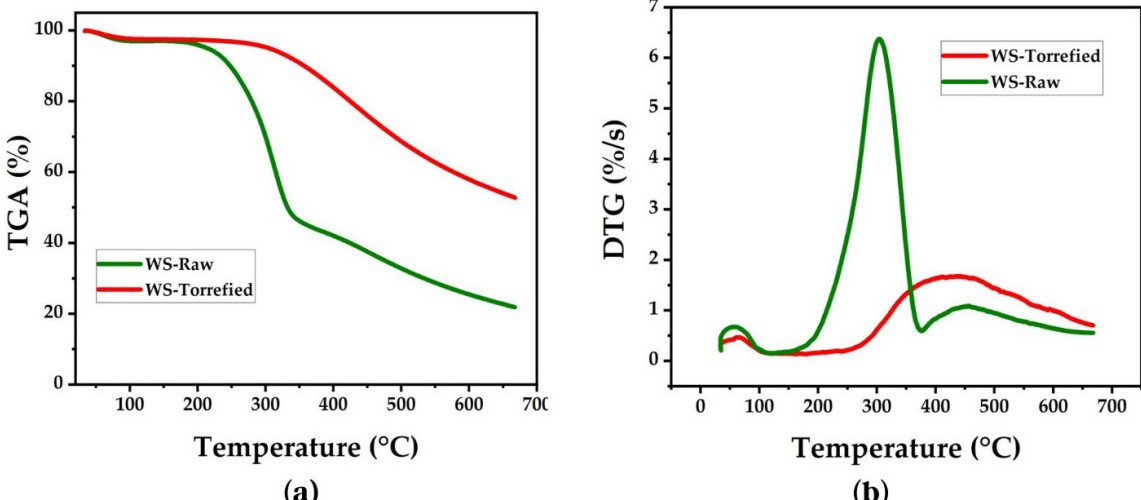

**Figure 4.** Curves (**a**) TGA and (**b**) DTG for wheat straw.

The DTG curves display the temperatures where the maximum mass loss occurs. In the untreated sample, the maximum thermal decomposition is observed at 300 °C, an intermediate temperature between the decomposition of hemicellulose and cellulose. The peak corresponding to lignin is observed at 450 °C. In the torrefied sample, the peaks of maximum mass loss have been shifted toward higher temperatures, close to 405 °C. When the temperature exceeds 500 °C, the DTG curves enter a zone of low reactivity. In this zone, the slow and progressive decomposition of carbonaceous residues formed in previous thermal stages takes place; similar results have been reported previously [39,49].

### 3.4.8. Moisture Absorption Analysis

The hygroscopic character of the plant biomass involves additional costs in transportation and storage and facilitates the biological degradation of the biomass. It was found that the optimal torrefaction conditions allowed a decrease in the moisture absorption capacity of wheat straw, changing the hygroscopic character to hydrophobic, as illustrated in Figure 5. Moisture absorption reduced from 125.82% in raw samples to 21.57% in torrefied samples, corresponding to a reduction in absorption of 104.25%. Similar research reported a diminution in moisture adsorption of up to 69% in wheat straw when microwave-assisted torrefaction was performed at 200 W–20 min [50]. In other residues, such as sugarcane bagasse, moisture adsorption reduced from 185.9% to 3.7% when torrefaction was performed at 270 °C–60 min [51]. The change in the hygroscopic character of biomass has been associated with eliminating functional groups that can form hydrogen bonds with water [52]. Among the functional groups that increase hygroscopicity is the hydroxyl group, which decomposes during torrefaction. The decrease of this functional group can be more evident in FTIR analysis.

### 3.4.9. FTIR Analysis

The chemical structure of the raw and torrefied biomass was examined using Fourier transform infrared spectroscopy. Figure 6 exhibits the absorption spectra of the samples in the region between 900 and 4000 cm$^{-1}$. The torrefied sample presented low transmittance compared with the raw sample. Previous research has reported on the degree of difficulty in obtaining high-quality spectra when very dark torrefied samples are used [53]. However, the spectra obtained allowed the observation of significant changes in the chemical structure of the biomass. These changes can be seen with the decrease and displacement of some characteristic absorption bands.

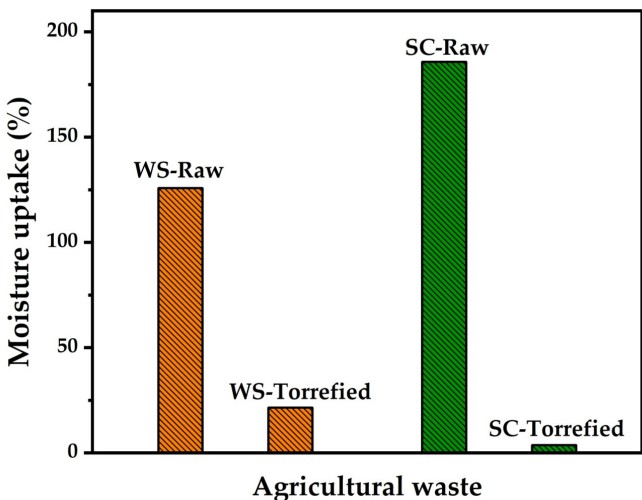

**Figure 5.** Moisture uptake of wheat straw compared with sugarcane (SC).

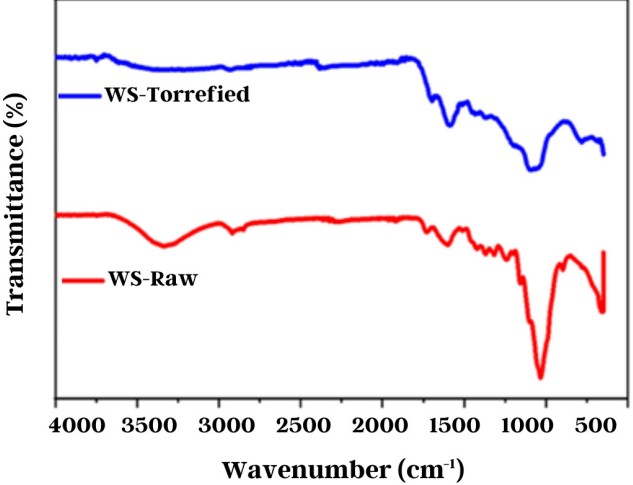

**Figure 6.** Wheat straw FTIR analysis.

After torrefaction, a significant decrease in the intensity of the O-H band around 3500–3100 cm$^{-1}$ was observed, which is attributed to the decline of the hygroscopic character of the torrefied biomass [53]. Additionally, other absorption bands were observed between 1700 and 1600 cm$^{-1}$, corresponding to the stretches and vibrations of the C=O and C=C groups of the lignin rings [46]. It was observed that these absorption bands shifted to lower wavenumbers after torrefaction, likely due to the formation of new compounds containing these functional groups [36,44]. Moreover, an absorption band of a great intensity of around 1100 cm$^{-1}$ was observed, corresponding to the vibrations of the C-O-C group of cellulose [54]. This band decreased in intensity after torrefaction, consistent with the experimental evidence reported in the literature [44,46,55].

### 3.4.10. SEM Analysis

Figure 7 exhibits the morphological and structural changes in the wheat straw before and after torrefaction. The lignocellulosic matrix and its fibrous structure can be observed in the untreated sample. In the torrefied sample, the biomass structure shows cracks and fissures. In addition, an increase in fiber separation and a considerable disintegration of the lignocellulosic components is observed. A comparison of the SEM images at 100× magnification indicates a substantial increase in the porosity of the residue and the formation of holes in the plant structure. The increase in porosity contributes to the decrease in the bulk density of the biomass, facilitating the densification process [8]. The generation of pores

and crack formation has been related to the release of volatile substances and the thermal decomposition of methoxyl groups present in the hemicellulose structure [43,56].

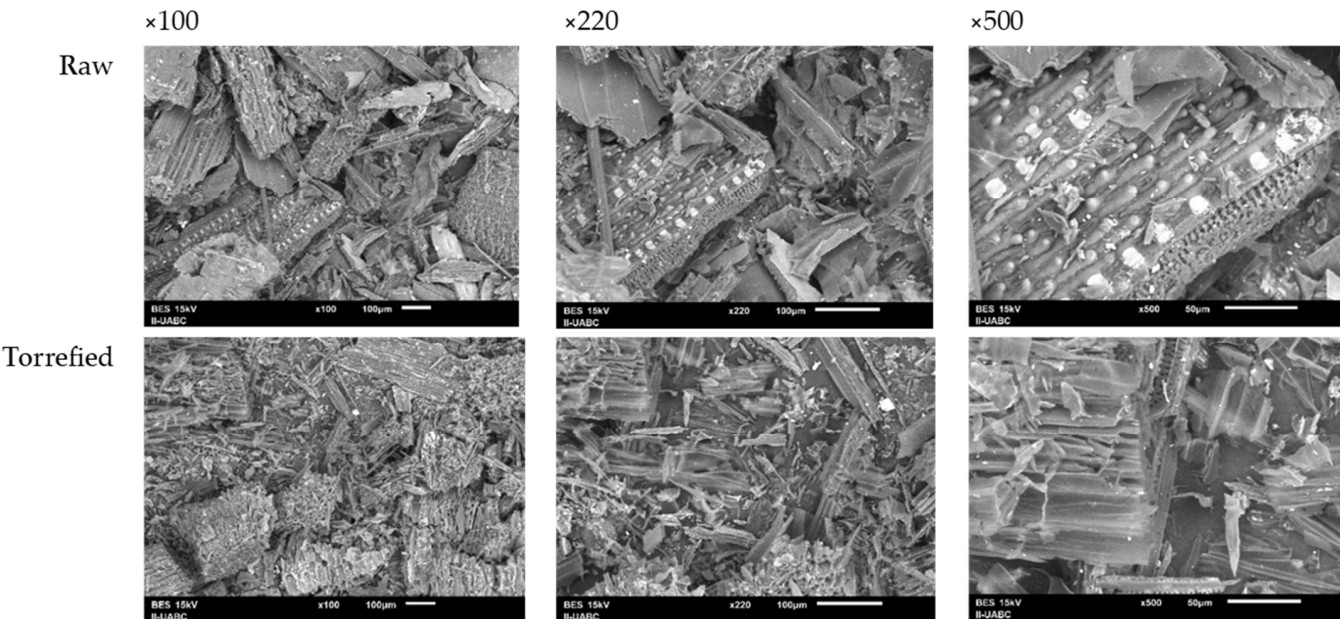

**Figure 7.** SEM images of wheat straw before and after torrefaction.

## 4. Conclusions

The effect of oxidative and non-oxidative torrefaction on torrefaction yields and fuel properties of wheat straw was studied. According to the statistical analysis, the temperature was the most significant factor in the process, while the reaction atmosphere was slightly more significant than the residence time. Oxidative and non-oxidative torrefaction allow the wheat straw to reach the HHV of type C sub-bituminous carbon. However, the oxidative condition enables this objective to be achieved with the least sacrifice in mass and energy yields. In addition, the HHV under oxidative conditions can be achieved with a lower requirement for retention time and reaction temperature, which could help to reduce operating costs.

The obtained mathematical models presented relatively high regression coefficients and adequately predicted the behavior of oxidative and non-oxidative torrefaction yields. The models allowed finding the optimal conditions for wheat straw torrefaction and achieving the energy value in the range of type C sub-bituminous char. In addition, these conditions allowed for improved physicochemical properties of wheat straw, which increased fixed carbon, elemental carbon, lignin content, and thermal stability. On the other hand, the content of volatile matter, moisture, hemicellulose, and elemental oxygen decreased. In addition, the number of hydroxyl groups in the chemical structure of wheat straw was reduced, which contributed to the decrease of its hygroscopic character. The plant structure of the wheat straw underwent considerable changes, specifically increased porosity, which is a factor that enhances the bulk density of the biomass. The torrefied biomass became a more fragile and brittle material, improving its crushability and grindability.

**Author Contributions:** Conceptualization, M.T.B.C. and R.T.R.; methodology, M.A.C.O. and M.A.C.Á.; software, R.T.R. and O.T.C.; validation, B.V.S., G.M.A. and M.T.B.C.; formal analysis, B.V.S. and M.T.B.C.; investigation, R.T.R. and M.T.B.C.; resources, M.A.C.O. and M.A.C.Á.; data curation, O.T.C. and M.A.C.Á.; writing—original draft preparation, M.T.B.C., R.T.R. and M.A.C.Á.; writing—review and editing, B.V.S., O.T.C., R.T.R. and G.M.A.; visualization, R.T.R. and M.A.C.O.; supervision, O.T.C., R.T.R., G.M.A. and M.T.B.C.; project administration, B.V.S., G.M.A. and M.T.B.C. All authors have read and agreed to the published version of the manuscript.

**Funding:** This research received no external funding.

**Data Availability Statement:** All data are available within the manuscript.

**Acknowledgments:** The authors gratefully acknowledge the support of Benjamín Rojano in this research.

**Conflicts of Interest:** The authors declare no conflict of interest.

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
