# Peer review of "Torrefaction under Different Reaction Atmospheres to Improve the Fuel Properties of Wheat Straw"

_processes, doi:10.3390/pr11071971_

Round 1
Reviewer 1 Report
The authors investigated the effect of oxidative and non-oxidative torrefaction on the torrefaction 10 yields and fuel properties of wheat straw. The factors temperature (230, 255, 280, 305 °C), residence time (20, 40, 60 min), reaction 12 atmosphere (0, 3, 6 vol % O2; N2 balance) were discussed. Compared to non-oxidative tor- 16 refaction, the results indicated a significant increase in HHV produced from oxidative torrefaction, 17 with substantial temperature and residence time reductions. The research topic is very interesting and the results are very important. Some useful data is given for readers and researchers. I think it can be accepted after minor revision. Some comments are as followed.
1 Table 5. Prediction of optimal conditions and torrefaction yields. Theoretical Conditions and Theoretical yields (%) of Type B and C are not listed. They are important.
2 in Figure 2. Surface graphics of response of temperature vs. atmosphere on the torrefaction yields of 267 the wheat straw for 40 min. The position of the coordinate axis is different. It is not clear for author comparison and analysis.
Author Response
The responses to reviewer 1 are included in the attached document.

Reviewer 2 Report
In the manuscript entitled "Torrefaction Under Different Reaction Atmospheres to Improve the Fuel Properties of Wheat Straw" the authors evaluated how the oxygen concentration in the reaction atmosphere reduces residence time, torrefaction temperature, and the production of solid fuel with similar properties to sub-bituminous coal without compromising the physicochemical properties of wheat straw.
Besides, the torrefaction process was optimized using RSM, and a systematic study of the fuel properties of raw and torrefied biomass was conducted. Overall, this manuscript is demonstrating an important and promising research direction, however part of this manuscript seems to be enhanced in much clearer discussion. Herein, it is suggested that the manuscript could be accepted for publication in Processes unless major corrections has been conducted according to the recommended points.
Required corrections:
1) Abstract: The authors should balance this section in both beginning and the end part, since it lacked of the innovation, importance and significance of this study, especially citing more numerical conclusions.
2) Key words: Optimization should be changed into Response Surface Methodology (RSM).
3) Introduction section, line 46-56: Since the topic was the potential aopplication of bimass convert to different solid fuels, especially the authors have mentioned the co-combustion for a better use of biomass and other materials, such as coal et al. However, some current studies about biomass pellets and bio-coal briquettes used in residential sector were missing, such as a new type of biocoal briquette, via a mixture of bituminous and biomass, which also showed an excellent performance in residential sector. It is suggested that the authors need to give a basic introduction and summaries in this part. Some recent literatures could be referred to: (a) Environmental Technology & Innovation, 2023, 29, 102975; (b) Bioresources Technology, 2021, 331, 124973.
4) In the Materials and Methods section, please supplement the statistical analysis, especially in the follwing tables, and indicate what is the software of data analysis in this study.
5) In the Results and discussion section: it is suggested that the authors should list the advantages and comparison of this study through comparing the current studies, and explain the reason. It is suggested that some recent literatures could be referred to: (a) Fuel, 2023, 336, 127166; (b) Case Studies in Thermal Engineering, 2022, 37, 102251.
6) The format of figure 3 and 5 should be adjusted and uniformed for clearly demonstration.
7) There are many format errors which suggested the authors need to be modified carefully throughout the whole manuscript.
8) The English language should be revised by an expert.
Moderate editing of English language required.
Author Response
The responses to reviewer 2 are included in the attached document.

Reviewer 3 Report
1. I think that the research on the use of wheat straw as a solid fuel for coal substitute through various torrefaction processes is a very important task.
2. In non-oxidative torrefaction environment, what is the reason why the change in PI-HHV value is less than in other conditions as the residence time increases only at the temperature of 280℃? (Page 6)
3. Why do energy yields increase significantly when the H/C ratio decreases in an oxidative torrefaction environment?
I think that the English writing ability of the thesis is at an average level.
Author Response
The responses to reviewer 3 are included in the attached document.

Round 2
Reviewer 2 Report
The revised manuscript is clear and significantly improved. Acceptance is recommended.